# Relative Influence of Land Use, Mosquito Abundance, and Bird Communities in Defining West Nile Virus Infection Rates in *Culex* Mosquito Populations

**DOI:** 10.3390/insects13090758

**Published:** 2022-08-23

**Authors:** James S. Adelman, Ryan E. Tokarz, Alec E. Euken, Eleanor N. Field, Marie C. Russell, Ryan C. Smith

**Affiliations:** 1Department of Natural Resource Ecology and Management, Iowa State University, Ames, IA 50011, USA; 2Department of Biological Sciences, The University of Memphis, Memphis, TN 38152, USA; 3Department of Entomology, Iowa State University, Ames, IA 50011, USA; 4Department of International and Global Health, Mercer University, Macon, GA 31207, USA

**Keywords:** West Nile virus, *Culex* mosquitoes, vector-borne disease ecology, mosquito surveillance, bird communities

## Abstract

**Simple Summary:**

West Nile virus (WNV) is transmitted by mosquitoes and maintained in bird populations. However, it is less clear why some geographic locations consistently serve as “hotspots” for increased WNV transmission. To address this question, we examined land use as well as mosquito and bird community metrics at sites across central Iowa. Our data suggest that WNV activity in mosquitoes is most heavily influenced by the abundance of *Culex pipiens* group mosquitoes during late summer, with landscape ecology having less defined impacts. Our data also suggest that bird community metrics have little influence on WNV infections in mosquitoes. Together, these results provide new information on the ecological and host factors that most heavily influence WNV transmission.

**Abstract:**

Since its introduction to North America in 1999, the West Nile virus (WNV) has resulted in over 50,000 human cases and 2400 deaths. WNV transmission is maintained via mosquito vectors and avian reservoir hosts, yet mosquito and avian infections are not uniform across ecological landscapes. As a result, it remains unclear whether the ecological communities of the vectors or reservoir hosts are more predictive of zoonotic risk at the microhabitat level. We examined this question in central Iowa, representative of the midwestern United States, across a land use gradient consisting of suburban interfaces with natural and agricultural habitats. At eight sites, we captured mosquito abundance data using New Jersey light traps and monitored bird communities using visual and auditory point count surveys. We found that the mosquito minimum infection rate (MIR) was better predicted by metrics of the mosquito community than metrics of the bird community, where sites with higher proportions of *Culex pipiens* group mosquitoes during late summer (after late July) showed higher MIRs. Bird community metrics did not significantly influence mosquito MIRs across sites. Together, these data suggest that the microhabitat suitability of *Culex* vector species is of greater importance than avian community composition in driving WNV infection dynamics at the urban and agricultural interface.

## 1. Introduction

West Nile virus (WNV) is the leading cause of mosquito-borne disease in the United States, causing more than 53,000 human cases and 2400 deaths since its introduction in 1999 [1]. WNV is maintained in an endemic transmission cycle involving *Culex* mosquito vectors and avian reservoirs, with human cases resulting from epizootic spillover [2,3]. Previous studies have implicated several environmental and ecological variables that drive seasonal patterns of WNV transmission [4,5,6,7]. In particular, land use and landscape ecology significantly impact vector and avian host communities [4,8,9,10,11,12,13,14], which ultimately shape WNV epidemiology at different ecological scales [12,13,15,16,17].

Regional variation in the distribution and abundance of *Culex* (Diptera: Culicidae) mosquito species, including *Culex pipiens*, *Cx. restuans*, *Cx. quinquefasciatus*, and *Cx. tarsalis*, influences regional differences in WNV transmission across the United States [4]. Moreover, the abundance of these principal WNV vectors can vary across land use gradients, where *Cx. pipiens, Cx. restuans,* and *Cx. quinquefasciatus* predominate in urban and suburban environments [4,18,19,20,21], while *Cx. tarsalis* is most abundant in rural and agricultural areas [4,13,22,23]. Therefore, land use and its subsequent impacts on vector ecology can profoundly influence local and regional WNV transmission dynamics.

Bird populations (primarily Passeriformes) also significantly contribute to WNV transmission, serving as the primary reservoir hosts for WNV infection [24] and promoting the dispersal of WNV through the movement of migratory birds [25,26,27]. While it has been suggested that overall species diversity in an avian community does not influence WNV transmission [28], previous studies suggest that mosquito infections are influenced by the abundance of nonpasserine bird species [28], as well as specific passerine bird species [16,29,30,31,32]. The American robin (*Turdus migratorius*) has often been implicated as a preferred host for *Culex* mosquitoes [16,29,30,31,32], and it has been suggested that robin migration drives seasonal shifts in mosquito host preference from birds to humans [29] or to other bird species [16,31]. Evidence suggests that these seasonal shifts to less competent avian hosts may further attenuate WNV amplification and spillover of WNV into human populations [16,17]. As a result, the abundance of particular avian species in a microhabitat may significantly contribute to the presence and transmission of WNV in specific bird communities.

Despite the well-described independent roles of the mosquito vector and avian host in WNV transmission, the relative contributions of mosquito and bird communities in defining mosquito WNV infection intensity across a regional landscape have not been adequately addressed. To approach this question, we examined mosquito surveillance data (2016 to 2018) and avian point counts (2018) at a series of sites across an ecological gradient in central Iowa, USA. For this region in which *Cx. pipiens* is believed to serve as the primary vector of WNV transmission [13,33], we demonstrate that “late-season” (after late July) abundance of *Culex pipiens* group mosquitoes is the primary driver of mosquito infection (a strong predictor of human cases [34,35]), with agricultural land use having a negative influence on mosquito infection rates. In contrast, bird communities had little effect on mosquito infection and did not display significant differences in their community composition across the ecological gradient employed in our study. Together, these data suggest that habitats most conducive to the expansion of *Culex* mosquito populations during peak times of WNV transmission represent the highest risk for the potential spillover of WNV into human populations.

## 2. Materials and Methods

### 2.1. Study Area

We examined eight sites in Polk and Story counties in central Iowa, USA (Figure 1). This included five sites in Polk County, which comprises the state’s largest city (Des Moines) and has the highest population density in the state. An additional three sites were examined in Story County within the city of Ames, the seventh most populous city in Iowa and home to Iowa State University. These trapping sites were included in the statewide WNV surveillance program conducted by Iowa State University and maintained by local public health partners in both Polk and Story counties. Sites from both counties share a similar ecology, average elevation (Polk, 919 ft; Story, 1017 ft) [36], and climate conditions [37].

### 2.2. Mosquito Trapping, Sample Identification, and WNV Testing

Mosquito collections were performed during the spring and summer of 2016–2018 from mid-May (epidemiological week 20) through the first week of October (epidemiological week 40). Mosquito abundance was determined using New Jersey light traps (NJLTs), while grass infusion-baited Frommer Updraft Gravid Traps targeted gravid adult female mosquitoes for subsequent WNV testing. The distance between traps varied by location but exceeded 30 m at each study site to not bias mosquito collections. Traps were run continuously throughout the trapping period (mid-May–October), with samples collected three times a week from both trap types. Gravid trap samples were immediately transported and stored at −80 °C for later identification, processing, and WNV testing.

Mosquito samples were identified according to morphological characteristics [38]. Due to the condition of samples from the NJLTs, the morphological features that distinguish adult female *Cx. pipiens* and *Cx. restuans* are often damaged in the collected samples [39]; thus, these species were collectively identified as “*Cx. pipiens* group” or CPG, as previously described [13,40,41]. While gravid trap samples better maintain these morphological features [33] due to the selective nature of gravid traps to select for certain *Culex* species [13], these were not included in our mosquito population analysis.

Following identification, *Culex* mosquito samples collected from gravid traps were assembled into pools of up to 50 specimens of the same species, site, and collection week, then sent to the State Hygienic Laboratory (Iowa City, IA) for WNV testing using detection by quantitative RT–PCR [42]. The presence or absence of WNV in mosquito pools from each trapping site location was used to calculate the minimum infection rate (MIR) for each site as previously described [13,43].

### 2.3. Bird Surveys

From 29 May–7 October 2018, a single observer (AEE) visited each site at least six times (seven visits to all sites except six visits to EWIN and YEBA). All visits occurred between the hours of 05:45 and 10:45 (between 15 min and 4.5 h after sunrise) to minimize variation due to time of day. At each site, the observer visited a series of points, organized in a hexagon, with edges of 100 m and including one point at the center near the mosquito trap for a total of seven survey points per site (Appendix A). All points were at least 100 m from one another. If a particular point of the hexagon fell at an unsafe location (e.g., the middle of a road or water body), its location was adjusted to be further than 100 m but still within 150 m from neighboring points. At all sites, safe locations were found for all seven points, with the exception of EMMC, at which only six points were surveyed due to the landscape. Within each site, the order in which points were visited was randomized for each trip. Upon arriving at each point, the observer stood quietly for 2 min, then recorded all birds seen and heard within the next 6 min, estimating their distances as 0–25 m, 25–50 m, or over 50 m, using a laser range finder (Aculon 6 × 20, Nikon, Melville, NY, USA). To minimize the chance that individuals were counted multiple times on the same day, our final analyses retained only birds detected within 50 m and excluded birds flying overhead. Surveys were only conducted in the absence of inclement weather (fog, steady drizzle, prolonged rain, wind speeds greater than 20 km/h, or lightning). These methods were adapted from the Iowa Department of Natural Resources Multiple Species Inventory and Monitoring Program [44], which was adapted from a similar program from the United States Forest Service [45].

### 2.4. Land Use/Land Cover Analysis

A high-quality (less than 10% cloud cover) Landsat 8 satellite image that encompassed all study sites was obtained using the public domain United States Geological Survey’s (USGS’s) EarthExplorer (https://earthexplorer.usgs.gov/, accessed on 18 Jul 2022). The image was imported into ArcGIS 10.4.1 software and edited using the buffer and clip tools to convert the full image file to an output extent of each study site [14,46]. The resulting output was composed of a circular image reflecting a 1 km radius surrounding each trap site. Land use/land cover was evaluated for each respective site by examining the following landscapes: barren land, water, agriculture/open, tree cover, and building/impervious, as previously described [14,46]. The percentage of each landscape was determined by using the number of pixels representing each land classification divided by the total pixel count of each site, with the resulting outputs converted to a percentage for each site.

### 2.5. Statistical Analyses

All analyses were performed in R version 4.1.3 (R Development Core Team, 2022) [47] using the packages ‘vegan’ [48], ‘ggplot2’ [49], and ‘MuMIn’ [50].

### 2.6. Mosquito Community Metrics

Since our mosquito surveillance data produced more frequent sampling than was feasible for birds, we divided each yearly mosquito data at the midpoint of our trapping season (corresponding to late July) into “early” (weeks 20–30) and “late” (weeks 31–40) trapping periods to examine temporal patterns in mosquito populations. These timepoints also denote important distinctions in historical WNV activity in Iowa, where 89% of human cases and 92% of WNV+ mosquito pools occurred between weeks 31 and 40 during the “late” trapping period [13]. For both trapping periods, as well as the entire season, we calculated the following metrics at each site, based on NJLT collections: total mosquitoes collected, total *Culex* spp. collected, total *Culex pipiens* group (CPG) collected, and percentages of both *Culex* spp. and CPG of the overall trap yield. To test how these metrics were related to MIR, averaged across years for each site, we calculated an early and late season average for each variable across 2016–2018. For the percentage of *Culex* spp. and CPG, we used averages weighted by the total number of mosquitoes captured. To describe intersite heterogeneity, we calculated the coefficient of variation (%CV) for each metric across sites (Appendix A), defined as the standard deviation divided by the mean, then multiplied by 100.

### 2.7. Avian Community Metrics

Within each site sampled in 2018, we calculated each species’ average number of detections per visit. Based on these numbers, we calculated the following metrics of avian alpha diversity for each site: mean detections per visit across all species, species richness (total number of species detected from May–October), Simpson’s diversity index [51], and Shannon’s diversity [52]. We also ran a principal component analysis (PCA) using data on all species to visualize differences in overall species composition among sites (beta diversity). In addition, we used amplification fraction estimates from Hamer et al. [31] to calculate a site-level WNV host competence index (HCI). Estimates of the species amplification factor were available for 24 of the 25 most commonly observed species in our dataset. Using information for these 24 species, for each site, we multiplied each species’ $\left(i\right)$ average detections per visit ($a_{i}$) by its estimated amplification fraction ($F_{i}$) and summed these such that:$$\mathrm{HCI}=\sum a_{i}\cdot F_{i}$$

For each metric of the avian community, with the exclusion of the PCA variables, we calculated the coefficient of variation across sites as above (Appendix A). Since PCA variables reflect a multivariate metric centered around a mean of 0, the equation for %CV cannot be interpreted meaningfully for these variables.

### 2.8. Relating Land Use and Community Metrics to WNV Prevalence in Mosquitoes

We used Akaike’s information criteria, adjusted for a small sample size (AICc) [53], to compare two sets of general linear models describing annual mosquito MIR. Both sets of models included a null (intercept-only) model.

The first set of models (16 total) fit the average annual MIR as a function of landscape characteristics (% forested/tree cover, % built, % agriculture, % water, % bare) at each site. To examine multivariate combinations, we examined models that included each landscape characteristic individually, as well as all possible pairwise summations of variables examined.

The second set of 96 models fit the average annual mosquito MIR as a function of up to two variables describing mosquito and avian communities. Each model included a maximum of one mosquito and one avian community metric. We chose the model with the lowest AICc value in each set as the best-supported model. We considered any models within 2 AICc units of the best-supported model as having substantial support [53].

## 3. Results

### 3.1. Study Sites, Land Use, and Mosquito Minimum Infection Rates

To examine the contributions of local avian and mosquito communities in defining differences in WNV prevalence across landscapes, we assessed bird and mosquito communities at eight sites in Story and Polk Counties in central Iowa, USA (Figure 1A). Together, these sites reflect diverse landscapes for the region, ranging in their ecology (Figure 1B). Moreover, these sites have displayed consistent differences in WNV activity from 2016–2018 (as measured by mosquito minimum infection rates, MIRs) (Figure 1A, Appendix A), suggesting that ecological differences between these habitats and their respective mosquito and bird communities may potentially influence their suitability for WNV transmission.

### 3.2. Mosquito Communities

To characterize how mosquito communities varied across our sites, we calculated the total number of mosquitoes collected (Appendix A), the percentage of *Culex* spp., and the percentage of *Culex pipiens* group (CPG) from New Jersey light traps (NJLTs) at each site location (Appendix A). Since the majority of WNV activity occurs in the latter half of the season (weeks 31–40) [13], we also examined our mosquito data temporally for the early (weeks 20–30) and late (weeks 31–40) parts of the surveillance season (Figure 2). Across the different sites, coefficients of variation for these mosquito community metrics averaged 69.2%, ranging from 26.6 to 114.7% (Appendix A). The majority of sites displayed large fluctuations in early-season mosquito numbers (Figure 2A), which were largely influenced by increased rainfall events and the emergence of *Aedes vexans* populations [41]. The percentage of *Culex* species and CPG also varied across sites in the early season, with sites showing distinct changes between early- and late-season time points (Figure 2B,C). Together, these data suggest that the mosquito communities varied extensively between sites and that *Culex* mosquito population dynamics varied temporally throughout the year across sites.

### 3.3. Bird Communities

To determine how avian communities differed across our sites, we calculated the overall abundance (average number of birds detected per visit; Figure 3A, Appendix A), species richness (total species detected; Figure 3B), and alpha diversity metrics (Simpson’s and Shannon’s diversity index; Appendix A). Although there were no clear differences in these community metrics between sites (Figure 3A,B), the abundance of specific bird species can distinguish sites by principal component analysis (Figure 3C). In addition, for each site, we calculated the host competence index [31], an indicator of how well a given bird community should serve as a functional WNV reservoir (Figure 3D). However, none of these community metrics displayed obvious associations with mosquito infections (MIRs) at our trapping locations (Figure 3). Metrics of avian communities were less variable across sites (CV range: 5.5–53.6%; Appendix A) than were metrics of mosquito communities (CV range: 26.6–114.7%, Appendix A), suggesting that bird communities are more uniform across site locations than *Culex* mosquito populations.

### 3.4. Ecological Factors and Mosquito MIR

Based on prior studies examining the association between land cover and WNV transmission by *Cx. pipiens* [54,55], we predicted that the minimum infection rate (MIR) of *Cx. pipiens* group mosquitoes would be highest at suburban sites with a high level of tree cover and impervious surface cover, representing ideal habitats for mosquito and bird species implicated in WNV transmission in central Iowa. To test this hypothesis, we used AICc to compare a series of linear models that predicted MIR by either individual landscape characteristics (measured within 1 km of the site) or pairwise combinations of these variables (Table 1 and Appendix A). Our most competitive variables had either positive or negative impacts on MIR values, with the percentage of tree and built landscapes having positive effects on MIR (Table 1), while the percentage of agriculture/open land paired with the percentage of water or bare soil having the largest negative impacts on MIR (Table 1, Figure 4A). These three variables were highly correlated with one another (|*r*| > 0.97 for all), suggesting that they all reflect similar characteristics of our sites. However, the intercept-only model had similarly high support in our analyses (ΔAIC = 0.75), suggesting that landscape characteristics were only weakly predictive of MIR (Table 1) and that other characteristics such as mosquito and bird communities may be more informative.

To test how our metrics of mosquito and bird communities were related to average annual MIR in mosquitoes, we again compared a series of linear models by AICc (Table 2 and Appendix A). The best-supported model contained a single predictor: the percentage of *Cx. pipiens* group mosquitoes collected during the late season (Table 2, Figure 4B). This variable also appeared in four of the five best-supported models (Table 2 and Appendix A). All models that included a metric of avian community composition showed less support (all ΔAICc ≥ 4.47, Table 2 and Appendix A), arguing that the dynamics of *Culex* mosquito populations alone are stronger determinants of WNV infection (MIR).

## 4. Discussion

Endemic WNV transmission is of significant public health concern in the United States, causing an estimated 7 million infections since its introduction [56]. While the separate roles of mosquito vectors and avian reservoir species in WNV transmission have been described previously, regional differences in ecology and species abundance have complicated our understanding of the ecological drivers of WNV transmission [4,55,57]. This is further supported by differences in vector ecology that shape WNV incidence in Iowa, where landscape and vector abundance drive WNV transmission dynamics across the state [13,23,33]. However, even at the county level, trapping site locations display consistent differences in WNV activity that further influence the dynamics of WNV transmission at the microscale. By examining the relative influence of land use, mosquito, and bird community composition on mosquito minimum infection rates (MIRs), our data suggest the increased abundance of *Cx. pipiens* group mosquitoes in the late summer (late July–October) most accurately predicted mosquito WNV infection at sites in central Iowa.

Unfortunately, due to the inability to morphologically distinguish *Cx. pipiens* and *Cx. restuans* in the New Jersey light trap samples collected in our study [13,40], we cannot definitively point to *Cx. pipiens* or *Cx. restuans* in driving these observed trends. Both species have previously been implicated as competent vectors of WNV and in amplifying WNV in reservoir bird populations [20,33,58]. However, several lines of evidence suggest that *Cx. pipiens* may have a more integral role than *Cx. restuans* in driving our observed MIR trends. *Cx. pipiens* consistently display higher MIRs than *Cx. restuans* [33], suggesting that *Cx. pipiens* is a more competent vector of WNV. This is further supported by the increase in WNV activity associated with the increased abundance of *Cx. pipiens* relative to *Cx. restuans* [33], and the more ornithophilic preferences of *Cx. pipiens* compared with *Cx. restuans* [13].

While temperature and rainfall undoubtedly shape *Culex* vector populations [33,59,60], it remains less clear as to the factors that define the differences in mosquito communities between our trapping locations. Oviposition behaviors and larval habitats of *Cx. pipiens* and *Cx. restuans* have not been clearly defined in the Midwest, yet are often influenced by vegetation, water quality, food resources, competition with other mosquito species, and the effects of predation [61,62,63]. Therefore, further efforts are required to study the factors that drive the habitat suitability of *Culex* species to better understand localized risks for WNV incidence.

Our analysis of avian host community metrics focused on passerine species abundance displayed little influence on mosquito infection rates. These findings are similar to previous studies in which passerine species richness did not influence WNV infection [28,64]. As a result, our data provide further support that passerine species abundance may not be a strong indicator of WNV transmission, as previously suggested [28,32], although nonpasserine species richness has been negatively correlated with human and mosquito infection rates [28]. Furthermore, when we examined our site locations according to the host competence index [31] as an additional metric to analyze the avian host composition, we found no relationship with mosquito infection rates. However, due to the strong feeding preferences of *Culex* mosquitoes for specific avian species such as the American robin (*Turdus migratorius*) [31,32], we cannot rule out that temporal abundance and susceptibility of a select number of species could help drive WNV dynamics within these avian communities.

One caveat of our analysis is the relatively uniform avian community structure across our sites in central Iowa. For example, approximately half of the observed bird species occurred at four or more sites, while a large colony of barn swallows at a single site (MOOR) contributed disproportionately to the observed differences in our PCA between sites. Therefore, the variation in avian communities captured in our central Iowa study locations may be substantially lower than in studies that concentrated on macroecological scales and found relationships between bird communities and vector-borne disease transmission [65,66]. However, our findings are consistent with previous work in central Iowa that found no difference in WNV seroprevalence between birds captured in agricultural versus urban landscapes [67].

Ecological differences in land use have also been implicated in shaping WNV transmission [7,23,54,55,57] due to their importance in defining mosquito and bird communities, as well as human population density and human activity in a given area. Similar to a previous study [54], we demonstrate that the mixture of impervious surfaces and green spaces often associated with suburban locations could enhance the potential for WNV transmission. In contrast, our predictive models, including the abundance of agricultural areas, display negative correlations with mosquito infection rates. These contrasting contributions are expected from peridomestic mosquito vectors, such as *Cx. pipiens* and *Cx. restuans*, which are associated with urbanized locations [20,68,69], and likely drive patterns of urban WNV transmission [17,54,70]. Within our geographically limited sample, however, these landscape features remained only weakly predictive of MIR (Table 1). Yet, at a larger ecological scale in Iowa and the greater Midwest/Great Plains region, where *Cx. tarsalis* drives WNV transmission in more rural areas, agricultural landscapes are important predictors of WNV cases in humans [4,13,23,57].

In summary, our study provides a novel evaluation of the relative influences of landscape, as well as mosquito and avian communities, on WNV infection rates in *Culex* mosquitoes. With mosquito MIRs serving as a strong predictor of human WNV cases [34,35], identifying the drivers behind these dynamics can have important public health implications for identifying potential hotspots and developing targeted interventions to reduce disease transmission.

## Figures and Tables

**Figure 1 insects-13-00758-f001:**
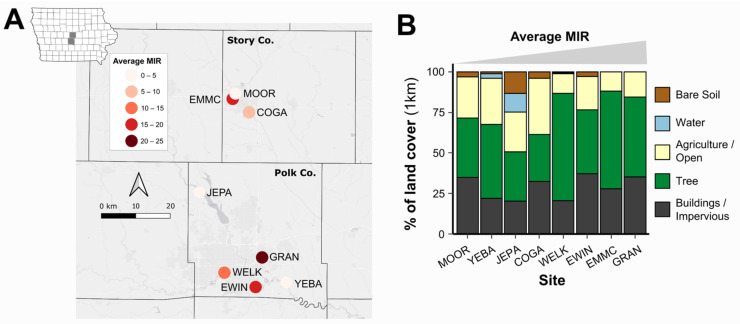
Minimum infection rates and landscape ecologies vary across mosquito trapping site locations. Study site locations in central Iowa showed substantial variation in mosquito minimum infection rates (MIRs, averaged from 2016–2018) (**A**) and displayed a wide range of landscapes, with land use classifications calculated by remote sensing within 1 km of trap sites (**B**).

**Figure 2 insects-13-00758-f002:**
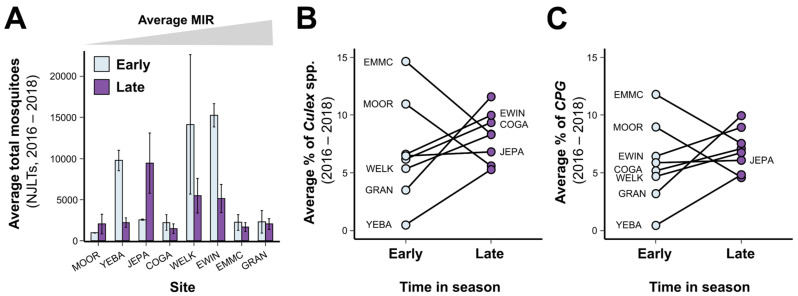
Mosquito communities vary across study sites and temporally during the season. Total mosquito numbers collected from NJLTs were averaged from 2016–2018 and displayed by site in order of increasing mosquito MIR (**A**). Data are displayed as the average ± SEM for the early (weeks 20–31) and late (weeks 31–40) times of the year. The average percentage of *Culex* spp. (**B**) or *Culex pipiens* group (CPG) mosquitoes (**C**) collected at each site compared with the total number of mosquitoes are displayed for both the early and late seasons. Connecting lines are to aid in the visualization of the data between time points.

**Figure 3 insects-13-00758-f003:**
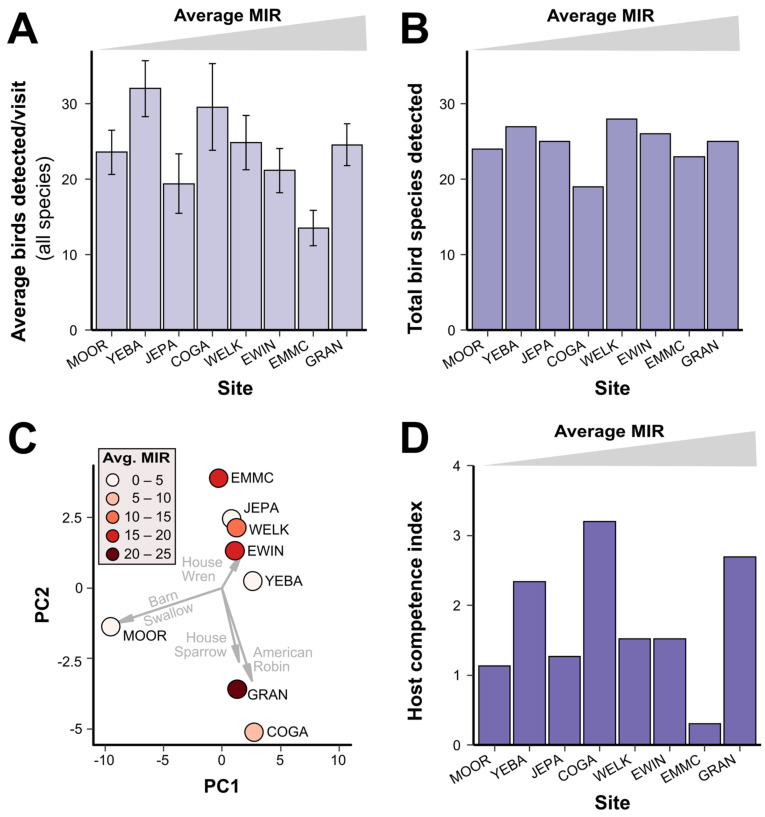
Avian communities are not indicative of mosquito WNV activity across sites. Descriptive plots of representative avian community metrics across study sites: abundance (**A**), richness (**B**), community composition visualized through principal component analysis (**C**), and host competence index (**D**). While all species were included in the analyses displayed in (**A**,**B**,**D**), only the four most common species from each site were used to generate the analyses depicted in (**C**). Common names of bird species most influential in the principal component analysis are displayed in gray.

**Figure 4 insects-13-00758-f004:**
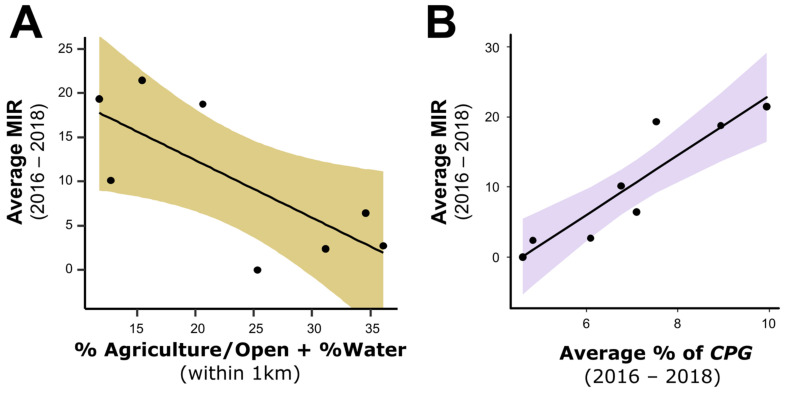
Predictors of mosquito MIRs. Correlations of mosquito minimum infection rate (MIR) with landscape or mosquito variables were examined among sites in central Iowa, USA. (**A**) The landscape variable that best predicted MIR was the combination of % agriculture/open plus % water (, ΔAICc = 0, adj. *r^2^* = 0.47, *F*_1,6_ = 7.27, *p* = 0.04). This variable correlated highly (*r* = −0.97) with the metric we predicted would best describe suburban habitats at high risk for WNV transmission (% tree plus % built). However, models including these variables showed only marginally better support than the null, intercept-only model (ΔAICc = 0.75). (**B**) Among metrics of mosquito and avian communities, the percentage of mosquitoes in the *Culex pipiens* group during the late season was the best predictor of MIR across sites (ΔAICc = 0, adj. *r^2^* = 0.81, *F*_1,6_ = 31.76, *p* = 0.001). Shading depicts ± 1 SE around model predictions.

**Table 1 insects-13-00758-t001:** Top five models predicting mosquito minimum infection rate (MIR)-based landscape characteristics.

Model	Predictor Variable	Slope	AICc	∆AICc	Weight	Adj. *r^2^*
1	% Agriculture/Open plus % Water	−0.65	61.73	0	0.22	0.47
2	% Agriculture/Open plus % Bare	−0.58	62.39	0.66	0.16	0.43
3	% Tree plus % Built	0.46	62.48	0.75	0.15	0.42
4	None (intercept only)	n/a	62.48	0.75	0.15	n/a
5	% Agriculture/Open	−0.72	63.25	1.52	0.10	0.36

For the model with no predictor variables (intercept only), *r^2^* (slope) and adjusted *r^2^* are not useful for comparison with other models, as both values will be 0 by definition. n/a, not applicable.

**Table 2 insects-13-00758-t002:** Top five models predicting mosquito minimum infection rate (MIR) based on metrics of mosquito and bird communities.

Model	Mosquito Variable	Mosquito Slope	Bird Variable	Bird Slope	AICc	∆AICc	Weight	Adj. r^2^
1	% *Cx. pipiens* group, Late Season	4.25	None	n/a	53.37	0	0.72	0.81
2	% *Cx. pipiens* group, Late Season	4.51	Cumulative Amplification Fraction	−2.53	57.84	4.47	0.08	0.87
3	% *Culex* spp., Late Season	3.33	None	n/a	58.31	4.94	0.06	0.65
4	% *Cx. pipiens* group, Late Season	4.44	PC2	−0.70	58.67	5.29	0.05	0.87
5	% *Cx. pipiens* group, Late Season	3.98	Avg. Total Detections per Visit	−0.32	60.31	6.94	0.02	0.84

n/a, not applicable.

## Data Availability

Mosquito NJLT abundance data are openly available at https://mosquito.ent.iastate.edu/ (accessed on 18 July 2022). Further mosquito gravid trap data or avian community counts are available upon request.

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
