# Peer review of "Relative Influence of Land Use, Mosquito Abundance, and Bird Communities in Defining West Nile Virus Infection Rates in Culex Mosquito Populations"

_insects, 2022, doi:10.3390/insects13090758_

Round 1
Reviewer 1 Report
Introductuion
the article refers to the Iowa surveillance program should be detailed in the introduction
line 46: the numbering of the bibliography should follow the order of appearance
line 76: better specify the sampling period (year ,day of start and end of monitoring)
line 75-84: they are part of the conclusions and not suitable for the introduction. authors should better define the aims of the study
Materials and Methods
Study area
better describe the study area by inserting a map in M&M and indicating altitude, meteorological data (average annual temperatures and rainfall etc..)
mosquitoes trapping
the paragraph should be improved with more information: distance between traps, time and time of positioning of the 2 traps, distance between the 2 different traps, how many samples per site. for culex not morphologically identified it would be advisable to carry out a species identification with biomolecular techniques, if it is not possible to define the reasons.
Birds survey
the paragraph should be improved with a better description of the site and observation methods by including a diagram describing the points chosen for observation and the times for each point
Avian Community Metrics
line171: better describe the indexes used by inserting the formula and the bibliographic reference
Results
the results should be expanded by referring to the different species of mosquitoes sampled not only Culex but also other genus in order to understand the sampling capacity of the traps. The species of birds observed and the number for each observation should be indicated
Discussion
Does the surveillance plan include WN tests on birds? in this case were they performed for the sampling sites? whether such data should be entered and related to the conclusions of the study.
Author Response
Reviewer #1
Introduction
The article refers to the Iowa surveillance program should be detailed in the introduction.
We would like to thank the reviewer for the suggestion, yet we disagree that additional details about the surveillance program (not already included in the methods section) should be included in the Introduction. We don’t believe that this is a key component of the research questions and narrative associated with the study, such that the inclusion of these details in the introduction would only add information that detract from the narrative of the introduction.
Line 46: the numbering of the bibliography should follow the order of appearance.
We appreciate the reviewer’s suggestion, although it is unclear what the reviewer means by “following the order of appearance”. We understood this comment as listing the references in chronological order, which has been corrected in the revised manuscript.
Line 76: better specify the sampling period (year, day of start and end of monitoring).
We would like to thank the reviewer for the comment. We have added the years of our mosquito and bird surveillance to the revised introduction. However, we did not provide the full details of our sampling period in the introduction, where we believe these additional details are not needed for this section of the manuscript. All of these details are included in our Methods.
Line 75-84: they are part of the conclusions and not suitable for the introduction. authors should better define the aims of the study
We would like to thank the reviewer for the comment. However, we believe that this is more a comment of our stylistic writing preference, that does not directly influence the presentation of our study. For this reason, we have chosen to leave this section of text unchanged. If the editor feels that these changes are indeed required to meet the requirements of the journal we are happy to modify.
Materials and Methods
Study area
Better describe the study area by inserting a map in M&M and indicating altitude, meteorological data (average annual temperatures and rainfall etc..)
We would like to thank the reviewer for the comment. We already have a map in Figure 1, such that an additional map in the methods, in our opinion, would be redundant. The study site locations in Polk and Story counties in Iowa represent a relatively homogenous landscape, with comparable elevation and climate conditions. In response to the reviewer, we have added additional text to our methods, better describing these study sites, as well as provide resources regarding elevation and meteorological data. However, since our study does not consider temperature/rainfall in our research questions, we do not provide specific data for our study locations since it does not influence the interpretation of the study. With the provided resources, this can now be further investigated by the reviewer or other readers if needed.
Mosquito trapping
The paragraph should be improved with more information: distance between traps, time and time of positioning of the 2 traps, distance between the 2 different traps, how many samples per site. for culex not morphologically identified it would be advisable to carry out a species identification with biomolecular techniques, if it is not possible to define the reasons.
We would like to thank the reviewer for the comment. In response to the reviewer, we have added additional detail to the methods to describe the distance between the trap types at each location in the revised Methods.
Both trap types were run continuously during our trapping period (May-October) and have added text to the revised Methods to better convey these trapping methods.
We have additional supplemental data (Table S4) that summarize the mosquito species collected at each site and year. As shown with these data, we collect several thousand Culex mosquitoes per year, such that molecular identification is impossible. Although imperfect, morphological identification is the only method to identify species and is widely utilized and adapted throughout the surveillance community. These challenges in morphological identification have already been described and referenced in the Methods section.
Birds survey
The paragraph should be improved with a better description of the site and observation methods by including a diagram describing the points chosen for observation and the times for each point
We would like to thank the reviewer for the comment. In response to the reviewer’s comment, we have added additional text to the Methods and have included a supplemental figure (Figure S1) to better illustrate the manner in which we performed our bird observations in our revised manuscript.
Avian Community Metrics
Line171: better describe the indexes used by inserting the formula and the bibliographic reference
We would like to thank the reviewer for the comment. Since measurements of diversity (Simpson’s/Shannon’s) are widely used in the ecology community, we don’t believe that it is necessary to include the formulas for each of the above equations. In our revised manuscript we provide the original reference for each method.
Results
The results should be expanded by referring to the different species of mosquitoes sampled not only Culex but also other genus in order to understand the sampling capacity of the traps. The species of birds observed and the number for each observation should be indicated
We would like to thank the reviewer for their suggestion. In our revised manuscript, we have included Table S4 which displays the total number of mosquitoes collected per site for each year of the study period. For the study area, the NJLT samples represent ~30 different mosquito species in varying abundance across the sample sites, demonstrating the wide utility of NJLT to capture estimates of the general mosquito population at a given location.
We have also included Table S6 and Table S7, which provides additional respectively for information to the number of observations for each bird species and a summary of observations at each of our study site locations.
These data are referenced in our revised results section.
Discussion
Does the surveillance plan include WN tests on birds? in this case were they performed for the sampling sites? whether such data should be entered and related to the conclusions of the study.
We would like to thank the reviewer for their comment. Wild bird populations have never directly been tested (alive or dead) as part of surveillance activities in Iowa. There has been one independent investigation examining WNV seroprevalence in bird populations in Iowa (Randall et al. 2013) which has already been included in the discussion.
Reviewer 2 Report
The manuscript "Relative influence of land use, mosquito abundance, and bird communities in defining West Nile virus infection rates in Culex mosquito populations" reports the analysis of field sampling of mosquitoes and bird community data from eight sites in Iowa, USA. It finds that mosquito abundance, particularly for Culex species, was a superior predictor of virus minimum infection rates that other factors (bird numbers, land use type). The article is well written, clearly presented and the conclusions are supported by the data. The following points must be addressed to complete the article:
1. Include a table in section 3.1 stating exactly what mosquito species were trapped and the exact minimum infection rate at each site at each time interval (early/late). It is not clear from the text or figures what is the actual data e.g. Fig 1A gives ranges for each site for the whole time span and Fig 1B does not have a scale associated with the MIR average.
2. The Figure legend for Fig 4 is identical to that for Fig 3, the correct legend should be included.
3. The authors should proof read the reference section and correct a number of errors and omissions: Ref 9, check format; Ref 10, incomplete & format; Ref 34, incomplete; Ref 41, is this article accessible by website?; Ref 45 looks incomplete; Ref 46 looks incomplete;
4. Revise sentence line 17 that begins "Our data also bird community metrics..."
Author Response
Reviewer #2
- Include a table in section 3.1 stating exactly what mosquito species were trapped and the exact minimum infection rate at each site at each time interval (early/late). It is not clear from the text or figures what is the actual data e.g. Fig 1A gives ranges for each site for the whole time span and Fig 1B does not have a scale associated with the MIR average.
We would like to thank the reviewer for their suggestion. We have added supplementary tables that summarize the species of mosquitoes trapped for each site/year (Table S4), the percentage of Culex species at each site (Table S5) and the range of MIR values for each site across years (Table S3) in our revised manuscript to address these suggestions.
- The Figure legend for Fig 4 is identical to that for Fig 3, the correct legend should be included.
We would like to thank the reviewer for catching this error in formatting. We apologize for the mistake and have corrected the figure legend in our revised manuscript.
- The authors should proof read the reference section and correct a number of errors and omissions: Ref 9, check format; Ref 10, incomplete & format; Ref 34, incomplete; Ref 41, is this article accessible by website?; Ref 45 looks incomplete; Ref 46 looks incomplete;
We would like to thank the reviewer for the comment. We apologize for the oversight. The references have been corrected in our revised manuscript.
- Revise sentence line 17 that begins "Our data also bird community metrics..."
We would like to thanks the reviewer for catching this omission. This sentence has been corrected in our revised manuscript.
Round 2
Reviewer 1 Report
No comments to add